# Water Appropriation on the Agricultural Frontier in Western Bahia and Its Contribution to Streamflow Reduction: Revisiting the Debate in the Brazilian Cerrado

**Andréa Leme da Silva** [1,*], **Saulo Aires de Souza** [2,3], **Osmar Coelho Filho** [2], **Ludivine Eloy** [4], **Yuri Botelho Salmona** [5] **and Carlos José Sousa Passos** [1]

1. Graduate Program for the Environment and Rural Development, Faculty UnB at Planaltina, University of Brasília, Vila Nossa Sra. de Fátima, Federal District, Brasília 73345-010, Brazil; cjpassos@unb.br
2. Graduate Program on Environmental Technology and Water Resources, Anexo SG-12, Térreo, Campus Darcy Ribeiro, University of Brasília, Federal District, Brasília 70910-900, Brazil; sauloaires@gmail.com (S.A.d.S.); mc2sustentavel@gmail.com (O.C.F.)
3. Brazilian National Water Agency (ANA), Setor Policial, Área 5, Quadra 3, Blocos "B", "L", "M" e "T", s/n, Federal District, Brasília 70610-200, Brazil
4. ART-Dev, Univ Montpellier, CIRAD, CNRS, Univ Paul Valéry Montpellier 3, Univ Perpignan Via Domitia, 34000 Montpellier, France; ludivine.eloy@univ-montp3.fr
5. Graduate Program on Forest Sciences, University of Brasília, Posto Ecológico, Federal District, Brasília 70297-400, Brazil; yuri@cerrados.org
* Correspondence: leme.andrea@gmail.com

**Abstract:** Over the last three decades, almost half of the Brazilian tropical savanna (Cerrado biome) has been converted into cropland and planted pastures. This study aims to understand the implications of the expansion of the agricultural frontier for water resources in western Bahia state. We use an interdisciplinary approach that combines quantitative and qualitative data (spatial and hydrological analysis, interviews) to tie together land use changes in the Corrente basin, the streamflow and precipitation time series in the Pratudão River sub-basin (part of the Corrente basin), and the perceptions of soybean farmers and smallholder communities about the transformations of the hydrological cycle over the last few years. We observed an almost 10-fold increase in agricultural surface area in the Corrente River basin over the last three decades (1986–2018), going on from 57,090 ha to 565,084 ha, while center-pivot irrigated areas increased from 240 ha to 43,631 ha. Over this period, the streamflow has reduced by 38% in the Pratudão River. Our hydrological analyses, based on the Mann-Kendall test, of seven fluviometric stations and 14 pluviometry stations showed a statistically significant streamflow trend in the Pratudão River sub-basin for both minimum and mean streamflow series ($p \leq 0.05$). Surface runoff coefficient, which relates streamflow and precipitation annual data coefficient, decreased from around 0.4 in the late 1990s to less than 0.2 in 2015. In addition, most precipitation time series analysis (number of annual rainy days) showed no statistically significant trend ($p > 0.05$). Our results indicate that agricultural changes rather than climate change may be the main driver of downward streamflow trends in the Pratudão River sub-basin that is part of Corrente River basin.

**Keywords:** hydrology analysis; agribusiness; smallholder communities; water appropriation; western Bahia; Corrente River basin; Brazilian Cerrado

## 1. Introduction

The fresh water "embodied" in a commodity, known as "virtual water," refers to the volume of water consumed or polluted ("grey virtual water") to produce a commodity and is measured over its full production chain. The use of surface water or groundwater ("blue virtual water") [1] have more than doubled over the last two decades, mainly due to exports of soybean, meat, and dairy products from Latin America to Asia and Europe [2].

The direct negatives effects of increased water use for export agriculture include water resource depletion, pollution, and concentration of water use rights, which can jeopardize local communities' livelihoods and the environment [1].

In Brazil, agriculture and livestock husbandry consume 78.3% of the water available in the country [3]. Since 2000, soybean cultivation has expanded thanks to increasing access to irrigation in the Cerrado biome. In 2018, Brazil produced 117.9 million tonnes of soybean, accounting for 42% of all soy exported globally [4] Center-pivot irrigated grain crops (e.g., cotton, beans, maize, and soy) occupy an area of 1.4 million ha in Brazil, and 80% of these pivots are concentrated in the Cerrado biome. This biome is of strategic importance to the Brazilian economy, thanks to the quantities of agricultural commodities produced there. The MATOPIBA region (acronym formed by the names of Maranhão, Tocantins, Piauí, and Bahia states) experienced a high growth in irrigated area, going from 9 center pivots in 1985 to 1762 center pivots in 2017 [3].

The Brazilian tropical savanna (Cerrado) is the second largest biome in South America, occupying approximately 2 million km$^2$. The Cerrado is well-known as a biodiversity hotspot [5] and is often referred to as "Brazil's water tank," supplying water to three important aquifers and six of the country's major water basins [6]. Some of the major rivers in Latin America, such as the São Francisco River, have their headwaters in the Cerrado biome.

Due to agricultural expansion, the Cerrado is most threatened biome in Brazil, with already half of it having been converted to cropland and planted pastures. From 1985 to 2017, more than 43% of the Cerrado biome's (86,264,156 ha) native vegetation was lost and now only 55% of the original vegetation remains [7]. In addition, only 8.3% of the remaining Cerrado is protected in conservation units, and an additional 4.2% in indigenous lands [8]. Most of these new agriculture frontiers have occupied native areas in the Cerrado biome or areas inhabited by peasants or traditional populations [9].

The conversion of the Cerrado's native vegetation is having significant impacts on ecosystem functioning, such as regional climate regulation, hydrological stability, and biogeochemical cycles, associated with the loss of significant carbon stocks and the replacement of biodiverse ecosystems [10,11]. Converting natural vegetation to agriculture substantially modifies evapotranspiration and streamflow in small catchments [12]. Moreover, removal of deep-rooted plants and trees from the Cerrado ecosystems reduces infiltration and recharge of aquifers, which affects this biome's water balance and hydrological cycle [6]. Some hydrological studies in the São Francisco watershed relate downward streamflow trends mainly to land use changes [13–16], while others attribute streamflow reduction mainly to climate change, represented by downward precipitation trends [17,18].

Despite evidence of aquifer overexploitation [19], the continued allocation of water grants to agricultural enterprises has been causing conflicts with groups of smallholders situated downstream from the plantations, especially during the dry season. The Correntina municipality (western Bahia state) is the most emblematic case of worsening water conflicts in this region. On 2 November 2017, a popular uprising, which came later to be known as the "Correntina water war," took place when around a thousand people occupied the farm headquarters of the Igarashi Company and destroyed installations used to pump water from the Corrente River to irrigation systems. The Bahia state environmental agency (Institute for Environment and Water Resources, INEMA) had earlier issued a 32-center-pivots water grant to the Igarashi group to pump water directly from the Arrojado River, which also supplies Correntina city. Between 500 and 1000 people invaded part of Igarashi's farm headquarters, and destroyed a significant portion of its facilities and equipment in order to protest water appropriation by agribusinesses. A week later, about 10,000 people marched through Correntina in defense of the Corrente River and its tributaries [20]. These actions took place during the dry season, when streamflow was very low, and after months of unsatisfied requests to the Corrente River basin committee for a temporary suspension of new water grants concessions until the watershed plan could be defined [20,21].

This article analyzes implications of the expansion of the agricultural frontier over water resources in western Bahia state with a focus on the Pratudão River sub-basin that is part of Corrente River basin. In order to understand changes in the hydrological cycle related to changes in land use across multiple scales, we adopted an interdisciplinary approach that combines quantitative hydrological dataset (precipitation and streamflow series), changes in land use across multiple scales, water use for irrigation (center-pivot irrigated areas), and interviews.

## 2. Material and Methods

### 2.1. Study Area

The western Bahia region includes three river basins, those of the Grande River, the Corrente River, and the northern part of the Cariranha River, all tributaries of the São Francisco River. The Corrente River basin has an area of 34,875 km$^2$ encompassing 11 municipalities (Brejolandia, Canápolis, Cocos, Coribe, Correntina, Jaborandi, Santa Maria da Vitória, Santana, São Felix do Coribe, Serra Dourada, and Tabocas do Brejo Velho) with a total population of 196,761 inhabitants [22]. In the Corrente River basin, we focused on the Pratudão River sub-basin because its unique landscape presents different ecosystems, peasant communities, large agricultural areas, a protected area and a history of water conflicts.

This region sits on top of the Urucuia aquifer system, a vast geological formation with an area of 76,000 km$^2$ that connects rivers and helps regulate their seasonality and interannual streamflow variability [23]. This aquifer is important for the maintenance of the São Francisco River basin, one of the most important of Brazil's river basins, known for the role it plays in linking several Brazilian states. The maintenance of this river basin takes place through rainwater infiltration on flat and elevated relief areas, where sandy latosols play a major role due to their porosity and permeability [24].

The regional climate is tropical humid (Aw according to the Koppen climate classification) [25] with two well-defined seasons: a rainy season from October to March, and a dry season from April to September. The average annual precipitation is between 500 and 1200 mm. However, rainfall is very irregular and the mean annual rainfall decreases sharply as one travels westwards. The rainfall is concentrated in the 100-km area along the Serra Geral mountain range (bordering Goiás state), where most agribusiness farms are located [23].

Initially occupied by indigenous peoples, the region was colonized by cattle ranchers in the 18th Century, which led to the formation of many villages along the rivers. Today the region has hundreds of indigenous, *quilombola* and traditional communities (*fundo* and *fechos de pasto*). They usually practice agriculture in the valleys (dominated by gallery forests and swampy forests), complemented by free-range cattle ranching and fruit gathering in collective lands located in dry and flat uplands (dominated by shrubby and open grasslands). Most of them use a traditional gravity irrigation system to cultivate land near homes: the canals, built at the beginning of the 20th century, to collect water from the springs and to irrigate faraway fields and gardens during the long dry period. This practice was responsible for self-sufficiency and prosperity of the region well-known by production of sugar, rice, beans, cassava flour, and meat [26,27].

The agro-industrial occupation of western Bahia state started in the early 1970s with eucalyptus and pine plantations by Bamerindus Bank, which established in the headwaters of the Pratudinho River and Pratudão watershed by obtaining a government concession. The agricultural settlement began in the early 1980s with first producers originated from southern Brazil (States of Rio Grande do Sul, Paraná, and São Paulo). Federal government incentives, such as low land prices, low interest rates, and planned infrastructure boosted migration processes [28].

Since the 1980s, smallholder communities have witnessed the deforestation and expropriation of most of their communal land due to agribusiness expansion, especially in the flat uplands, which have been converted into monocultures. In addition to the decline

in cattle ranching due to the loss of rangelands, last years they saw the rivers' springs and irrigation canals dry up [26,29].

### 2.2. *Corrente River Basin Precipitation and Runoff Time Series Analysis*

The methods to assess the runoff and precipitation in the Corrente River basin included hydrometeorological data collection, exploratory data analysis, and trend detection. For the hydrological pattern analyses, daily precipitation and streamflow data were obtained from the National Water Agency (ANA) Hydrological Information System (Hidroweb). In order to have the highest quality with the longest available daily data, we selected all stations in operation that have data records that go back at least 30 years (Figure 1).

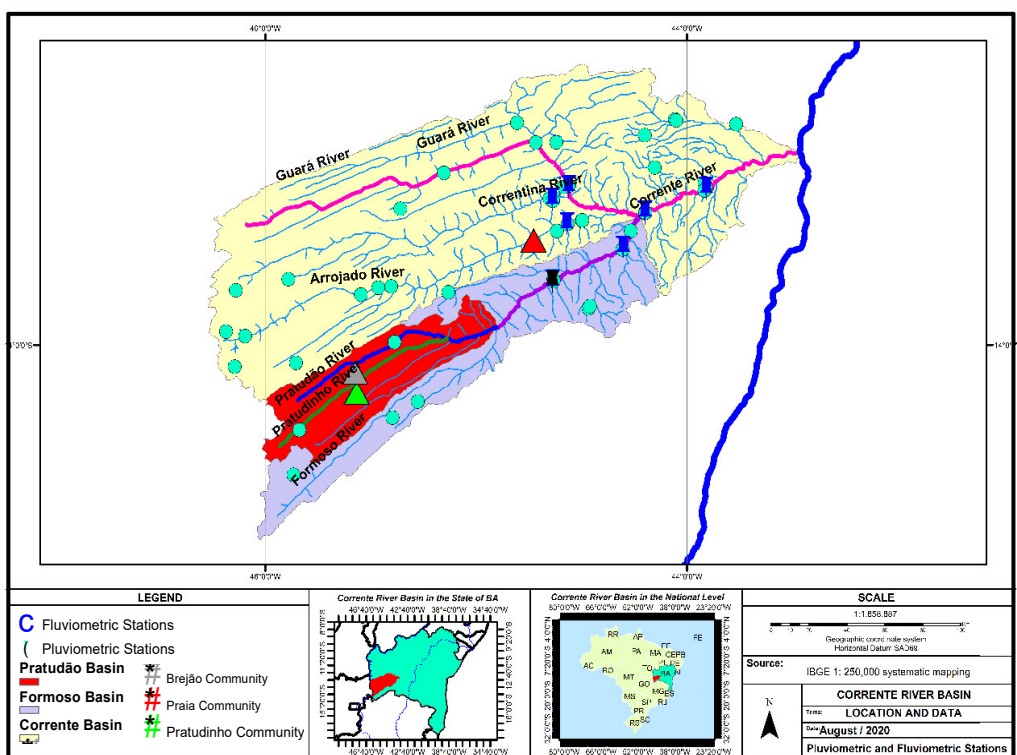

**Figure 1.** Location of the fluviometric (pushpins) and pluviometric (circles) stations in the study area. Black station corresponds to fluviometric station 45840000 (ANA Station Code) (the only one located near the outlet of the Pratudão River sub-basin). Source: Hydrological Information System of the Brazilian National Water Agency (Hidroweb).

Exploratory data analysis (EDA) was used to assess problems and inconsistencies in the dataset. Data were carefully examined for quality and an exhaustive data quality control was conducted in order to identify spurious errors and inconsistencies on dataset. We used the traditional technique of double mass curve [30]. Additionally, graphs were used to explore, understand and present mean, maximum, and minimum precipitation and streamflow. To assess the presence of outliers we used a generalization of the Grubbs test [31,32] which usually detected outliers either too small or large. In this study, it was not necessary to use any procedure to fill gaps in time series.

Possible precipitation and streamflow changes were evaluated using the Mann-Kendall test for trends (MK), utilizing daily, monthly, and yearly series of mean, maximum, and minimum values from both the rainy season and the dry season. We adopted the "Trend Free Pre-Whitening" (TFPW) approach to consider the effect of autocorrelation in time series data. The following section better explains the methods used to detect trends in a time series with significant serial correlation. For the analyses of rainfall patterns in the Corrente River basin, we considered 14 rainfall stations with at least 30 years of data without any failure (Table 1). For the streamflow patterns, evaluation was carried out by

considering, in addition to the series of streamflow in the Pratudão watershed, the seven fluviometric stations with at least 30 years of data without any failure located across the Corrente River basin (blue fluviometric stations in Figure 1 and Table 2).

**Table 1.** Selected rainfall stations shown in Figure 1 (n = number of years without failures).

| ANA Station Code | Station Name | Latitude | Longitude | First Year | End Year | n |
|---|---|---|---|---|---|---|
| 1344017 | Santa Maria Da Vitória | −13.40 | −44.19 | 1945 | 2019 | 68 |
| 1344001 | Coribe (Rio Alegre) | −13.83 | −44.46 | 1936 | 2000 | 40 |
| 1344013 | Gatos | −13.71 | −44.63 | 1952 | 2019 | 56 |
| 1244018 | Santana (Santana Do Brejo) | −12.98 | −44.05 | 1936 | 2000 | 55 |
| 1344015 | Colônia do Formoso | −13.56 | −44.30 | 1962 | 2019 | 47 |
| 1344002 | Mocambo | −13.27 | −44.55 | 1945 | 2019 | 70 |
| 1344008 | Vila de Acudina | −13.19 | −44.15 | 1964 | 2000 | 34 |
| 1344004 | Correntina | −13.33 | −44.63 | 1936 | 1985 | 47 |
| 1344016 | Arrojado | −13.45 | −44.56 | 1977 | 2019 | 43 |
| 1343008 | Porto Novo | −13.29 | −43.90 | 1936 | 2019 | 79 |
| 1344007 | Santa Maria da Vitória | −13.40 | −44.2 | 1939 | 1977 | 33 |
| 1344014 | Correntina | −13.33 | −44.65 | 1972 | 2019 | 43 |
| 1344010 | Santa Maria da Vitória | −13.40 | −44.2 | 1919 | 2012 | 45 |
| 1344009 | São Sebastião dos Gatos | −13.70 | −44.63 | 1936 | 1976 | 38 |

**Table 2.** Selected streamflow stations (n = number of years without failures).

| ANA Station Code | Station Name | River | Latitude | Longitude | Drainage Area (km²) | First Year | End Year | n |
|---|---|---|---|---|---|---|---|---|
| 45590000 | Correntina | Correntina | −13.34 | −44.63 | 3900 | 1977 | 2019 | 43 |
| 45740001 | Mocambo | Do Meio | −13.28 | −44.56 | 7950 | 1977 | 2019 | 43 |
| 45880000 | Colônia do Formoso | Formoso | −13.55 | −44.30 | 9550 | 1972 | 2019 | 43 |
| 45840000 | Gatos | Formoso | −13.71 | −44.63 | 7130 | 1952 | 2019 | 41 |
| 45960001 | Porto Novo | Corrente | −13.29 | −43.90 | 31,000 | 1977 | 2019 | 41 |
| 45770000 | Arrojado | Arrojado | −13.45 | −44.56 | 5540 | 1977 | 2019 | 43 |
| 45910001 | Santa Maria da Vitória | Corrente | −13.40 | −44.19 | 29,600 | 1977 | 2019 | 42 |

Streamflow and precipitation time series were obtained through regionalization and interpolation techniques. In order to assess the conditions for the hydrometeorology changes according to its population perception, interviews with local communities and farmers were carried out in the Pratudão River sub-basin.

The drainage-area ratio method was commonly used to estimate streamflow for sites where no streamflow data were available. It utilized data from one or more nearby streamflow-gauging stations. The method was intuitive and straightforward to implement, and its use was widespread among surface-water resources managers and analysts [33–36]. The method estimated the streamflow at an ungauged location by multiplying the measured flow at the nearby reference gauge by the area ratio between ungauged watershed and the gauged one [34]:

$$Q_{ungauged} = Q_{gauged} \frac{A_{ungauged}}{A_{gauged}} \tag{1}$$

in which $Q$ represents streamflow and $A$ represents watershed area. A major assumption of the area ratio method is that streamflow scales directly with watershed area. Thus, as watershed area increased, streamflow rate increased at some fixed rate per unit area. The choice of reference gauge in the area ratio method had generally been determined by geographic proximity to the ungauged watershed of interest [36,37]. We followed this suggestion, using the Gatos gauge (code 45840000) in the Formoso river as a reference gauge. The Pratudão River sub-basin covered more than 50% of this reference gauge

drainage area, which made this gauge suitable to be the data donor to apply regionalization procedures.

To assess changes in rainfall-runoff behavior, we used the evolution of runoff coefficient (RC) in the Pratudão River sub-basin. The annual RC was a good indicator for monitoring change over time in a catchment. An increase or decrease in this indicator over time would indicate possible land degradation events [38–40]. The runoff coefficient was defined as the ratio between runoff observed at the hydrological station (in this study Pratudão River sub-basin) and the spatially averaged rainfall over the basin. RC was analyzed at annual time scales. In order to obtain the RC, it was necessary to transform the streamflow unit from m$^3$/s to mm/year by dividing the streamflow by the drainage area of the Pratudão river sub-basin. The average precipitation was based on inverse interpolation distance weighted (IDW) by using the data from the Corrente River basin rainfall stations.

### 2.2.1. Mann-Kendall Trend Test

The rank-based Mann-Kendall trend test [41,42] was adopted in this study to identify statistically significant increasing or decreasing monotonic trends. The Mann-Kendall test was based on a non-parametric approach, which does not require the normal distribution of the data. The test was also applicable to data containing outliers or non-linear trends [43,44]. The Mann-Kendall test was used because it is distribution-free, robust against outliers, and it has been applied to non-normally distributed data (Yue, Pilon, and Cavadias 2002a). In addition, it has been used in most previous streamflow trend analyses [45,46]. If $x_1$, $x_2$, $x_3$, ... , $x_n$ is the time series of length $n$, then the Mann-Kendall test statistic $S$ is given by:

$$S = \sum_{i=1}^{n-1} \sum_{j=i+1}^{n} sign(x_j - x_i) \tag{2}$$

where:

$$sign(x_j - x_i) = \begin{cases} 1, if\ x_j > x_i \\ 0, if\ x_j = x_i \\ -1,\ if\ x_j < x_i \end{cases} \tag{3}$$

The null hypothesis $H_0$ for the test is "there is no trend in the time series." If $H_0$ is true, then $S$ is normally distributed with a mean of zero and variance:

$$\sigma_0^2 = \frac{n(n-1)(2n+5) - \sum_{j=1}^{m} t_j j(j-1)(2j+5)}{18} \tag{4}$$

where $n$ is the number of data points, $m$ is the number of tied groups and $t_i$ is the number of ties of extent $i$. A tied group is a set of sample data having the same value. In cases where the sample size $n$ is greater than 10, the standard normal test statistic $Z$ is computed as:

$$Z = \begin{cases} \frac{S-1}{\sigma}, if\ S > 0 \\ 0 \quad, if\ S = 0 \\ \frac{S+1}{\sigma}, if\ S < 0 \end{cases} \tag{5}$$

The sign of $Z$ indicates the trend in the data series, with positive $Z$ values indicating an increasing trend, and negative $Z$ values indicating decreasing trends. For the tests at a specific $\alpha$ significance level, if $Z < Z_{\alpha/2}$ or $Z > Z_{1-\alpha/2}$, the null hypothesis is rejected, and the time series has a statistically significant trend. In this study, the significance level $\alpha$ was 5%.

### 2.2.2. Theil-Sen Slope Estimator

The magnitude of trends has been determined using the Theil-Sen approach (TSA) [47]. The non-parametric procedure for estimating the slope of the trend in the sample of $N$ pairs of data is given by:

$$d_i = \frac{(x_j - x_k)}{(j - k)}, \; for \; i = 1, \ldots, N \tag{6}$$

where $x_j$ and $x_k$ are the data values at times $j$ and $k$ ($j > k$), respectively. The TSA slope $\beta$ is given by:

$$\beta = med\{d_i\} \tag{7}$$

### 2.2.3. Autocorrelation

The Mann-Kendall test required the input data to be serial independent. The presence of positive serial correlation in the data structure can lead to an overestimation of the trends' significance [48]. As it was shown by studies on streamflow [45,49] and precipitation [50,51], hydrometerological data are typically autocorrelated. To overcome the effect of serial correlation, the Trend-Free-Pre-Whitening (TFPW) technique was adopted in this study.

The prewhitened (PW) approach was developed to correct an inflated or deflated type I error to a preassigned significance level [48]. However, it also reduced the test sharpness. Yue et al., Rivard and Vigneault, and Wang et al. [44,52,53] found out a disadvantage arises from the deflationary trend slope in prewhitened series, and suggested estimating the slope before whitening the series in a process known as trend-free prewhitening (TFPW). Thus, the selection of TFPW procedure was due to its ability to deal with the presence of serial correlation, avoiding excessive decrease in rejection rates, that is, avoiding to obscure some trends that actually are significant. In this procedure, the trend slope was first estimated, and the record was detrended. Then the detrended series lag-1 serial correlation coefficient was estimated, and the series were pre-whitened using this estimate. Finally, the identified trend was added to the pre-whitened series. Then the Mann-Kendall test was applied to this series to assess the trend significance. Yue et al. [44] argued that trend removal as a first step may allow for a more accurate estimation of the population's lag-1 autocorrelation coefficient, and a subsequent better estimation of the trend significance.

The analysis of daily precipitation changes was undertaken by considering the fact that an annual rainfall change can be caused by the occurrence, combined or not, of two phenomena: fewer rainy days over the year, and lower rainfall on each of the rainy days over the year. This analysis was essential to evaluate the degree of influence of climate factors, in addition to land use change patterns, on the Corrente River basin's hydrological regime, and its influence in the Pratudão River sub-basin.

### 2.3. Changes in Land Cover Use and Intensification of Water Use for Large-Scale Irrigated Agriculture

In order to assess the history of Corrente River basin land use, specifically in the Pratudão River sub-basin, the data from MapBiomas version 4 (2019) which covered the period from 1985 to 2018 were cut and grouped in categories (e.g., native vegetation, pasture, agriculture, and others) through a script generated by Google Earth Engine. Center-pivot irrigated areas was obtained from geospacial data in the National Water Agency homepage (https://metadados.snirh.gov.br/geonetwork/srv/por/catalog.search#/home, accessed on 30 November 2020).

### 2.4. Interviews

In order to evaluate the local communities' perceptions of the impacts of large-scale agriculture, we observed productive activities and technologies on site, 20 semi-structured interviews were conducted in three smallholder communities (Brejão, Pratudinho, and Praia) between 2017 and 2019. The interviewees were selected based on specific criteria: smallholders that live from small-scale agriculture who were native or had lived in the region for at least 15 years. Most of these people were descendants of immigrants originally

from the northeast (Bahia state) or from the northwest (Goiás state) who settled in this region in the early 20th Century to escape not only from slavery-like labour, but also from drought and infertile lands. Currently, about 650 residents (210 families) live in Brejão and 53 residents in Pratudinho (10 families). The Praia community, consisting of 30 families, was located on the Arrojado River, another tributary of the Corrente River. The Pratudinho and Brejão communities were located in a wildlife refuge protected area (Refúgio de Vida Silvestre or REVIS), the Veredas do Oeste Baiano. This REVIS was located in the municipalities of Cocos and Jaborandi, in the western region of Bahia state and covered an area of 128,050 ha. It was created in 2002 (Federal decree of 13 December 2002) to protect the Pratudinho and Pratudão River springs, both of them tributaries of the Formoso River. This wildlife refuge was the only protected area focused on the water resources conservation in western Bahia state, and it had suffered with increasing water demand by irrigation use in its buffer zone.

We also carried out interviews with seven agribusiness stakeholders; three of them having water grants for irrigation issued by INEMA. Two of these interviewees pumped water directly from the Formoso River (12,410 ha of the irrigated area), and the third one pumped groundwater in the Pratudinho River headwaters (2565 ha of the irrigated area) inside the Pratudão River sub-basin. During the interviews, we asked questions about production patterns, and changes in water flows, as well as their possible or hypothetical causes and consequences. Concerning the number of interviews, we emphasize that the sample size is very small and may not be representative of all large-scale farmers.

In a previous publication [54] (further information available on the studied farms, cultivated varieties, productivity per unit area, irrigation systems used, etc.) noted these same farms had been operating, on average, for 14 years (ranging from 5 to 30 years). The total size of rural properties corresponded to 165,280 ha, ranging from 1650 to 80,000 ha (average of 10,000 ha). Soybean farms rotated the soybean crop with corn, cotton, and beans. In addition to grains (Pro, RR and conventional), farms equipped with center pivot irrigation produced soybean and transgenic maize seeds, based on trade agreements with Monsanto Dow, Pioneer, Brasmak, and Embrapa. One farm also cultivated tobacco in partnership with Phillip Morris. Four farms without an irrigation system also cultivated brachial seeds, native beans, and sorghum, which constituted plant species more adapted to drought.

## 3. Results

### 3.1. Corrente River Basin Rainfall and Discharge Trends

The Mann-Kendall test (MK) results for the streamflow and rainfall series are summarized in Figure 2a–e and Tables 3 and 4. These results analyses were performed considering the statistical significance ($p < 0.05$) and the magnitude of change (*bsenrel*), evaluated in terms of the percentage of change over a decade related to the series average.

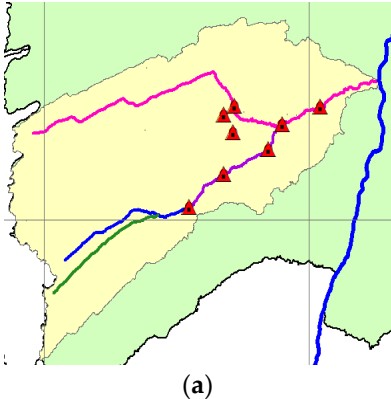　　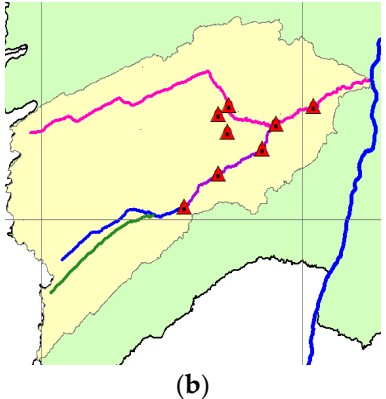　　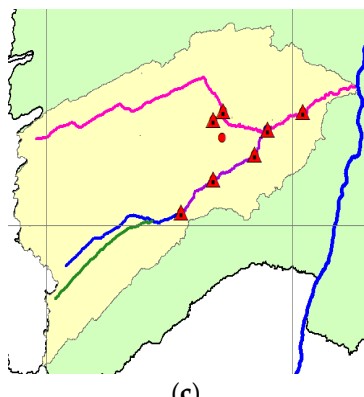

(**a**)　　　　　　　　　　　　　(**b**)　　　　　　　　　　　　　(**c**)

**Figure 2.** *Cont.*

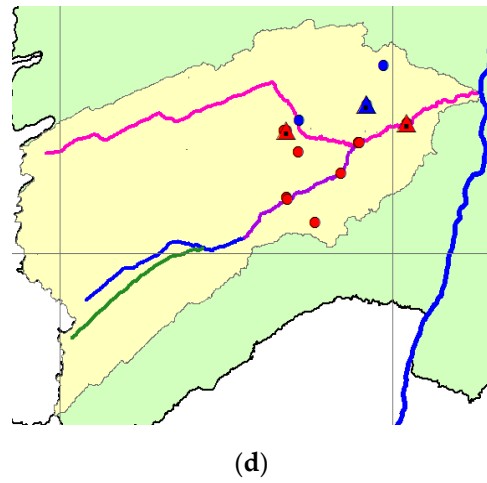

(**d**)

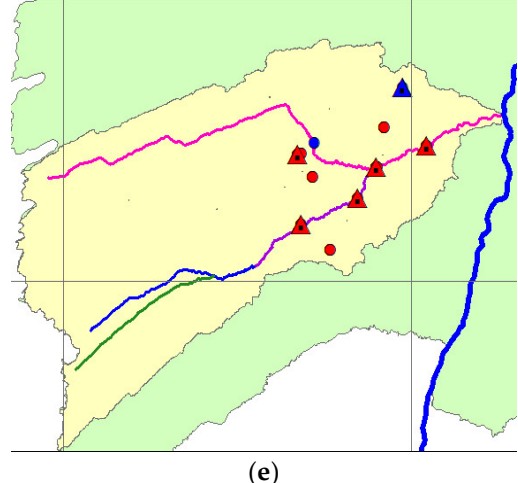

(**e**)

**Figure 2.** Trend analysis for streamflow and precipitation in the Pratudão watershed. (**a**) Trend analysis of minimum streamflows (*Qmin*), (**b**) trend analysis of mean streamflows (*Qmean*), (**c**) trend analysis of maximum streamflows (*Qmax*), (**d**) trend analysis of total annual precipitated volume (*PRCPTOT*), and (**e**) trend analysis of the number of annual days with precipitation (*ndays*). Legend: Red Circle: downward trend (statistically not significant); Blue Circle: upward trend (statistically not significant); Red Triangle: statistically significant downward trends at 5% error level; Blue Triangle: significant upward trends at 5% error level. The initial and final year of the evaluated period is shown in Tables 1 and 2.

**Table 3.** Statistical significance ($p < 0.05$) and the magnitude of change (bsenrel) of streamflows indices evaluated in terms of the percentage of change over a decade (%/Dec). The initial and final year of the evaluated period is shown in Tables 1 and 2.

| ANA Station Code | Qmin | | Qmean | | Qmax | |
|---|---|---|---|---|---|---|
| | *p*-Value | Bsenrel (%/Dec) | *p*-Value | Bsenrel (%/Dec) | *p*-Value | Bsenrel (%/Dec) |
| 45880000 | $3.77 \times 10^{-15}$ | −18 | $2.74 \times 10^{-13}$ | −17 | $2.13 \times 10^{-2}$ | −7 |
| 45960001 | $6.95 \times 10^{-13}$ | −17 | $2.67 \times 10^{-10}$ | −16 | $1.20 \times 10^{-3}$ | −11 |
| 45590000 | $9.57 \times 10^{-14}$ | −10 | $1.89 \times 10^{-12}$ | −11 | $4.57 \times 10^{-5}$ | −9 |
| 45840000 | $1.94 \times 10^{-13}$ | −16 | $6.05 \times 10^{-12}$ | −15 | $3.40 \times 10^{-7}$ | −14 |
| 45740001 | $5.73 \times 10^{-14}$ | −17 | $6.76 \times 10^{-12}$ | −17 | $9.12 \times 10^{-4}$ | −14 |
| 45770000 | $1.74 \times 10^{-13}$ | −12 | $1.03 \times 10^{-10}$ | −11 | $2.04 \times 10^{-1}$ | −5 |
| 45910001 | $4.44 \times 10^{-15}$ | −17 | $4.54 \times 10^{-12}$ | −16 | $1.87 \times 10^{-3}$ | −13 |

All fluviometric stations (Figure 2a–c, and Table 3) show a statistically significant downward trend for both minimum and mean streamflow series ($p \leq 0.05$). For maximum flows, just one station's null hypothesis trend was not rejected ($p > 0.05$), however, this station has shown a downward trend. These results indicate a decreasing streamflow trend, markedly in the minimum flows. The change magnitude was always less than −10%, reaching −18% for *Qmin* in the station 45880000, which means the minimum streamflow decrease rate in that station was −18% every ten years.

Figure 2d,e, and Table 4 illustrate the rainfall time series results, and the number of annual rainy days (*ndays*). Using data from 14 precipitation stations, most precipitation series showed no significant trends ($p > 0.05$). Only four stations had a significant decreasing trend and one station showed a significant upward trend ($p \leq 0.05$) (Figure 2d). It is important to note the only station 1344008 time series, which had a statistically significant upward trend ending in 2000, indicated the existence of possible subtrend resulting from interdecadal variability or long-term persistence. This result has raised the question whether the currently observed trends cannot have a long-term variability component not considered in the trend statistical tests.

**Table 4.** Statistical significance (*p*-value < 0.05) and the magnitude of change (bsenrel) of rainfall indices evaluated in terms of the percentage of change over a decade (%/Dec), total annual precipitated volume (PRCPTOT). The initial and final year of the evaluated period is shown in Tables 1 and 2.

| ANA Station Code | PRCPTOT | | ndays | |
|---|---|---|---|---|
| | *p*-Value | Bsenrel (%/Dec) | *p*-Value | Bsenrel (%/Dec) |
| 1244018 | 0.938 | 0 | 0.021 | 4 |
| 1343008 | 0.001 | −5 | 0.006 | −3 |
| 1344001 | 0.329 | −10 | 0.121 | −12 |
| 1344002 | 0.857 | 0 | 0.132 | 2 |
| 1344004 | 0.034 | −9 | 0.261 | −3 |
| 1344007 | 0.189 | −6 | 0.010 | −8 |
| 1344008 | 0.005 | 13 | 0.255 | −4 |
| 1344009 | 0.436 | −3 | 0.002 | −9 |
| 1344010 | 0.577 | −1 | 0.131 | −3 |
| 1344013 | 0.721 | −1 | 0.124 | −2 |
| 1344014 | 0.012 | −8 | 0.047 | −6 |
| 1344015 | 0.092 | −4 | 0.007 | −9 |
| 1344016 | 0.012 | −8 | 0.233 | −3 |
| 1344017 | 0.325 | −1 | 0.875 | 0 |

Annual rainy-day presented similar results, which reinforced our hypothesis that climatic factors were not the only drivers of hydrological regime changes in the Pratudão River sub-basin. These results suggest that climatic factors connected to rainfall have not played a major role in explaining the streamflow decrease shown by the runoff coefficient. This means that other factors associated with land use might be causing streamflow reduction.

The rainfall variability trends are among the most important issues in climate change research. Figure 3 shows the Pratudão River mean discharges (red line) and interannual rates of precipitation (inverted blue bars) from 1940 to 2016. The dashed black line superimposed on the graph represents their respective trend lines. Figure 3a–c show that total rainfall pattern does not exhibit any trend behavior over time, while there is a strong downward trend in the series of annual discharges since the 1980s for the mean, maximum and minimum streamflows. The lowest values for average flows have been observed after 2000.

These two dimensions were compared by the runoff coefficient (RC) that established the relationship between streamflow and the means of annual rainfall. In the late 1990s, RC became less than 0.4, which means that of the total precipitated water less than 40% was converted to surface runoff (Figure 4). In 2015, RC dropped to less than 0.2. Such a dramatic drop during this period coincides with an intensification of soybean production based on irrigation in the studied region.

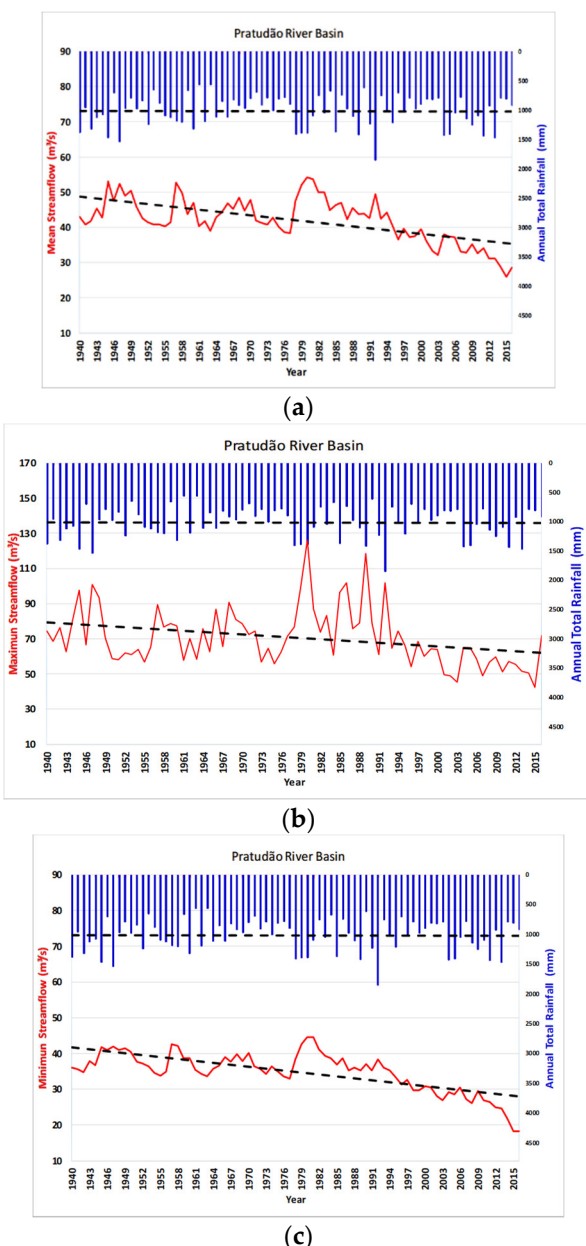

**Figure 3.** Mean (**a**), maximum (**b**), and minimum (**c**) annual streamflow series and annual total precipitation in the Pratudão River basin (1940 to 2016).

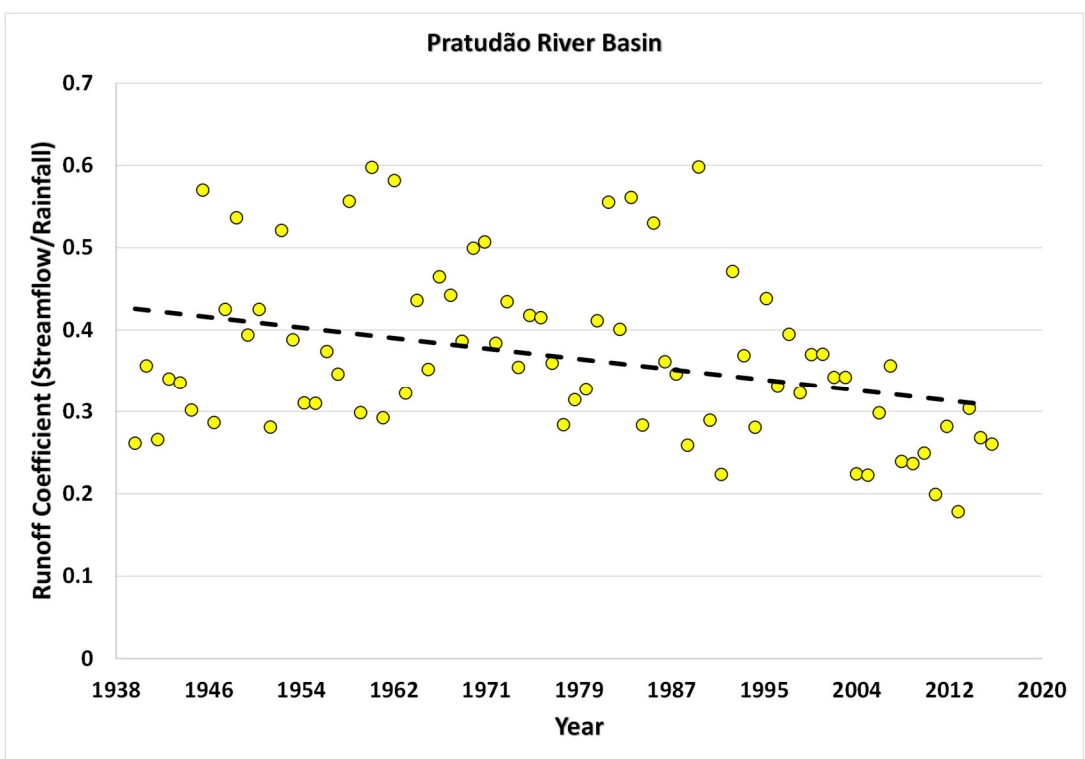

**Figure 4.** Runoff coefficient in the Pratudão river basin from 1940 to 2016.

*3.2. Intensification of Water Use for Large-Scale Irrigated Agriculture in the Corrente River Watershed*

Based on MapBiomas land use changes and land cover dataset to the Corrente River basin, we note that the agricultural area has grown from 57,090 hectares in 1986 to 565,084 hectares in 2018, a 10-fold increase over the analyzed period. In the Pratudão River drainage area, intensive agriculture increased from 14,392 hectares in 1986 to 62,491 hectares in 2018 (Figure 5).

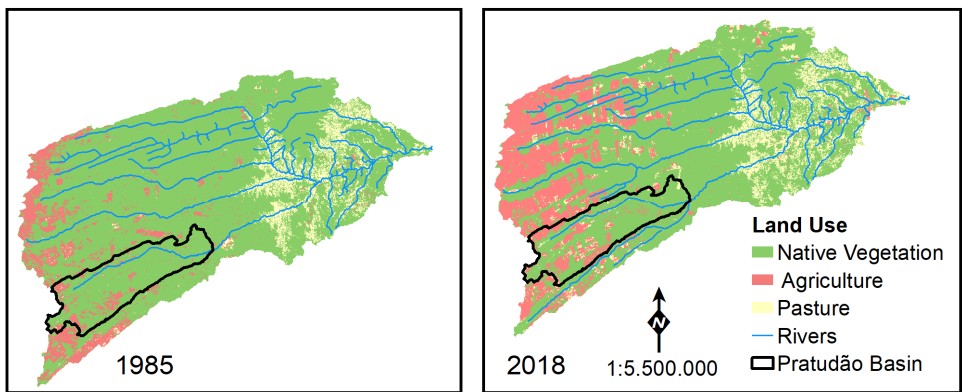

**Figure 5.** Land use change in the Corrente River basin (Pratudão sub-basin highlighted in black). Source: MapBiomas. Available from: https://mapbiomas.org/colecoes-mapbiomas-1?cama_set_language=pt-BR (accessed on 30 November 2020).

In the past three decades, the water flow has decreased sharply at the Pratudão River proxy station (data transferred from the station 4584000). While the 1985's streamflow rate was 388 mm, it dropped to 239 mm in 2016, a 38% reduction over the entire period.

Center-pivot irrigated areas increased from 240 ha to 43,631 ha in the Corrente River basin between 1985 and 2017. In Jaborandi, Cocos, and Correntina municipalities, where

most central pivots are placed, irrigated areas increased from 5965 ha to 31,164 ha between 2000 and 2017 (an increase of 522%). Moreover, we observe a concentration of center pivots mainly in the aquifer catchment areas of the Formoso, Arrojado, and Corrente Rivers.

*3.3. Conflicts Related to Water Access in Western Bahia*

The smallholder communities of the Pratudão watershed have reported a reduction in river flows after the agribusiness boom of the 2000s. About 90% of the interviewees blamed the reduction in the availability of river water flow on center pivots (both surface and groundwater abstraction). Around 60% pointed out a greater irregularity of precipitation patterns and/or rainfall reduction. About 90% of peasants observed a significant reduction in rainfall over recent years, which they attribute primarily to the expansion of agribusiness on the plateaus (catchment areas). They drew attention to the drying of veredas, which are areas with widespread occurrence of *buriti* tree (*Mauritia flexuosa*), a species that is an indicator of swampy areas.

About 25% of interviewees associated the reduction in river water volumes with deforestation. They also associated rainfall regime changes with vegetation suppression, longer and unforeseeable summers, along with more sparse rains. For instance, the women observed that the places on the river shores (as marked by the stones) where they used to wash clothes are getting more and more distant from the waterline.

The interviewees informed us that large-scale deforestation and irrigation cause a phenomenon of "spring migration" from the upper to the lower lands, followed by the gradual disappearance of smaller tributaries. For example, residents identified eight streams that have dried up in the Arrojado River valley. As a result, it is almost no longer possible to cultivate rice in these valleys, as was the case 10 years ago.

In addition, field observations have shown that this process of drying up is gradually disrupting the farming systems of traditional communities even in territories more than 50 km downstream from plantations, evidence of the large-scale impact of agro-industrial operations. Most swampy valleys (*veredas*) in which these communities traditionally cultivated rice began to dry out about 15 years ago and, more recently, centennial irrigation canals are drying out as well. This process has resulted in a decrease in agricultural output from properties that are no longer under watering systems, as well as led to conflicts between smallholders on the issue of water sharing. This situation is engendering resident unrest as Bahia state's Institute for Environment and Water resources, INEMA, keeps issuing water grants. The consequence is the intensification of water conflicts in western Bahia state over water grants for irrigation in the catchment areas (e.g., "water war" in 2017).

The reduction in river flows, as also the drying up of some springs and lakes, seems to be caused by the intensification of large-scale irrigated agriculture in the headwaters of the Formoso, Pratudão, and Pratudinho Rivers. The interviewees also noted the decreasing river waterflows, and the disappearance of springs and lagoons, such as Bamerindus lagoon (lagoa Bamerindus or Lagoa Feia), with the expansion of eucalyptus plantations in the 1970s as well as increasing number of artesian wells for animal husbandry after the 2000s. One interviewee reported that:

> " . . . these waters have fallen too low since 2000 . . . you could see the lagoons from far way, (like) Lagoa Feia [springs of Pratudinho], Pratudão . . . I was a child when I met that lagoon that was lost in sight, the view that there were people from a sea . . . you could see springs on the road . . . we used to go there by boat . . . today it's a dustbowl" (Peasant, Brejão, 2017).

In contrast, all interviewed soybean producers associate the reduction in the Formoso River waterflow to the reduction in precipitation in the region (e.g., longer summer periods and less or unpredictable rainfall), instead of the intensification of water use by irrigation. The naturalization of broader processes such as climate change and the greenhouse effect by these large farmers allows them to disclaim responsibility for the depletion of water resources associated with large-scale irrigation. As an example, one of the interviewed

farmers considers the drying up of Bamerindus lagoon (or Lagoa Feia) as a "natural process," which has also happened to other lagoons in the region. On the other hand, soybean producers focus consensually on "development" and "Cerrado's natural vocation" as narratives for food production that justify land and water use. They have also claimed that exports of grain and meat contribute significantly to the national gross domestic product (GDP), which are crucial to global food security. At the same time, about 60% of interviewees complain about excessive regulatory bureaucratic requirements concerning environmental and labor matters.

## 4. Discussion

### 4.1. Hydrological Analyses and Intense Irrigation Growth

Our results suggest that the decrease in streamflow cannot be attributed solely to climatic factors, which means that other factors associated with land use might be responsible for reductions in streamflows. Several studies have already examined changes in indicators based on daily precipitation and streamflow in the São Francisco watershed, which includes the Corrente River basin [13,14,17,55,56]. Most of the findings about changes in indicators are supported by this current analysis, which spans more recent years, and which uses the best available data for change analysis. Most studies have noted mainly downward trends, for both precipitation and streamflow, changes associated with drier conditions. The results for streamflows, unlike those for precipitation (significant results were always less than 15% of stations used), were significant for the most part (the significant results were always greater than 70% of the stations used). Moreover, recent studies on the evolution of the Urucuia Aquifer also have shown a significant decrease of its groundwater level since year 2000 [19,57,58]. As example, decrease in terrestrial water storage (TWS) (Terrestrial water storage (TWS) can be defined as the sum of all water on the earth's surface and in the subsoil. Composed of the root zone soil water storage (SWS) and groundwater storage (GWS). Satellite image processing (GRACE method) allows continuous monitoring of the TWS.) was 6.5 $\pm$ 2.6 mmyr—1 between 2002 and 2014, representing a total water loss of 9.75 km$^3$ (out of 125,000 km$^2$) on the surface of the Urucuia aquifer system [58].

It is important to emphasize that we have not underestimated the issue of climate change, but comparing the weights given to this driver. Considering the hydrometeorological variables trend analysis, our results are similar to other studies, where change in precipitation were few or almost nonexistent [13,14,17,59]. As an example, Pousa et al. (2019) found that only a small area of the Corrente River headwaters presented significant change of precipitation ($p < 0.05$). This still unclear change sign in precipitation disallow to consider climate change as the main driver of streamflow downward trend.

A relevant point in the results presented in Pousa et al. [17] is that short- and long-term persistence in hydrometeorological time series was not considered. The hydrometeorological variables have a propensity to be presented in clusters during certain periods of time, i.e., droughts or floods, that is termed 'scaling' or 'persistence'. Short-term persistence (STP), the most common and simple example, has been addressed in many studies using the autoregressive-1 model. On the other hand, long-term persistence (LTP) indicates that the process is compatible with the presence of fluctuations on a range of timescales, which may reflect the long-term variability of several factors such as solar forcing, volcanic activity and so forth [60]. The emergence of persistence may produce some intense local slopes, which might be falsely identified as deterministic trends but actually are only segments of large-scale random fluctuations. As widely reported in the literature [49,60–63], the presence of STP or LTP features considerably increase the chance of type I error, leading to an overestimation of the rate of significant results in statistical hypothesis tests used to detect changes [61]. These LTP features have already been reported in the São Francisco River basin hydrologic series [15,55,64]. Additionally, the statistically-relevant results for precipitation series in the station 1344008 from 1964 to 2000, and the station 1244018 from 1936 to 2000 (Table 4), suggest the presence of subtrends that are characteristic of the existence of long-term persistence [65,66].

Given that climate change and human activity are the two major drivers that can alter hydrological cycle processes, all these studies attribute these two factors as important drivers of the changes observed. However, other than Pousa et al. [17], these studies did not assess which of these drivers plays a more prominent role. These authors used a 36-year period (1980–2015) time series derived from a grid obtained by interpolating the rainfall stations in the region [67], with time series size smaller than those existing in pluviometric stations, which is the main feature in detecting changes [68]. Furthermore, the use of grid data necessarily implies the presence of spatial correlation between the grid data. This fact along with temporal autocorrelation increases the probability of type 1 error [49,69]. All of the above-mentioned issues have increased the uncertainty in the relationship between precipitation and streamflow drivers of change.

We believe that without in-depth knowledge and objective analysis about changes in the rainfall-runoff process, it is not possible to make such an attribution. It is a fact that the last two decades have been drier in Corrente basin than earlier, and climate projections for the region seem to agree that future conditions will be even drier [54,70]. However, currently, with few statistically-relevant results in precipitation, the drier climate in this period seems to be more associated with long-term natural variability (LTP) than nonstationary processes. In short, it is possible to have drier periods without having a statistically-relevant downward precipitation trend. For example, the only two significant upward trend in the rainfall indices (Figure 2d,e) come from series ending in the year 2000 (Table 1). It is likely that if the observed records (these stations are no longer in operation) continued the result would be different given that the following years were much drier. This suggests that these recorded data could be a small segment of a longer cycle of natural processes that, under current circumstances, are unidentified by currently available data. The fact that this downward trend is not significant only reinforces the idea that changes in flows have a greater chance of being associated with anthropological changes in land use. It should be noted that human activity in the basin does not lead only to increased water withdrawal, as suggested by Pousa et al. [17], but also alters the entire dynamic of the rainfall-runoff process, interfering in the hydrological regime, including in the wet period. Thus, the fact that the Flow Duration Curve (FDC) between two different periods are parallel may not necessarily result only from climatic factors since the difference can also be explained by changes in rainfall-runoff behavior as a whole. Furthermore, it is aggravated in the low-flow part where the magnitude of withdrawals can play a significant role, which explains a larger gap in the lower part of the FDC. We suggest that human activity plays a role in the observed streamflow changes as, according to Figure 4 (Runoff Coefficient), for the same total annual precipitation, the streamflow generation process has been smaller, indicating that even if the wetter period observed between 1981 and 1992 returns, the streamflow will possibly still be lower due to the change in rainfall-runoff behavior, which our study suggests is due to the replacement of native vegetation by irrigated agricultural areas in this region. But other interventions, with minor contribution in terms of occupied areas, such as eucalyptus planting, may also play a role in altering the water regime, as reported in other studies [71–73]. Similarly, according to [57,58], water storage depletion in the Urucuia aquifer is driven by anthropogenic impacts rather than by natural climatic variability.

In Brazil, few studies have evaluated the connections between land use changes and climate variability (e.g., river flows and rainfall), especially in the Cerrado. Such studies could help us understand the impacts of agribusiness and large-scale irrigation on streamflows. According to our knowledge, Dorneles, Souza, and Reis-Jr [74] were the first to objectively separate the impacts of climate change and human activities on streamflow in the São Francisco River basin (including the Corrente River basin). Heterogeneous results were obtained regarding the origin of the contributions within the basin. With 78% of the stations analyzed, both climate and anthropic activities contributed to the reduction observed in the flows.

Trends in water balance components across the Brazilian Cerrado based on precipitation, evapotranspiration, and terrestrial water storage can be useful to evaluate dry periods and assess changes in the water balance due to land cover and land use change [75] Therefore, changes in river basin hydrological patterns may reflect the increase in irrigation in recent years, as well as changes in land use (e.g., deforestation).

*4.2. Intensification of Irrigation Water Use by Large-Scale Agriculture and Social Conflicts*

Our results show that large-scale irrigation technologies can intensify social conflicts between farmers and local communities by changing the territorial dynamics (e.g., migration), since pivot farms impact areas where traditional communities and peasants have made a living. In western Bahia, where 90% of pivots of the MATOPIBA region are located, social conflicts highlight the current problem between irrigated industrial agricultural systems and the right to water [29].

Water conflicts have been reported in different regions of the Cerrado biome, a result of the rapid advance of agribusiness [29,54]. According to the Land Pastoral Commission, large-scale irrigated agriculture is one of the main causes associated with conflicts in Brazil. Between 2005 and 2017, the number of conflicts over water increased from 71 to 197, involving 35,418 peasants, indigenous and traditional peoples. In 2019, Bahia state alone recorded 56 incidents related to irrigation, including the water conflict in the Correntina municipality [20].

On the other hand, the irrigators' association from Bahia State (Associação dos Produtores e Irrigantes da Bahia—AIBA), a powerful organization that is representative of the large producers in western Bahia, focuses on "environmental sustainability", "social responsibility", and "sustainable growth" of large-scale irrigated agriculture, glossing over water scarcity, social conflicts, and water crises. Lately, since the potential for surface water use seems to be reaching its limit, the AIBA narrative has begun to focus on the "groundwater potential" and "water monitoring" of the Urucuia aquifer [76]. In addition, the narrative of the Cerrado's "natural vocation" to produce food shared by soybean producers and some agri-environmentalists [77] is coupled with a narrative of a "natural process" of decreasing water availability caused by climate change [18]. Both narratives reinforce each other and leave little room for critical knowledge on the role of industrial agriculture in decreasing water availability in the Corrente River basin.

Climate change can be an intensifier of decreasing streamflow trends in the Pratudão watershed, driven by large-scale irrigated agriculture in the medium and long term, according to a set of projections of climate models [16,70]. Therefore, water conflicts in Bahia state are a result of a water governance system based on an ineffective participation process informed by incomplete hydrometeorological data and weak law enforcement [78]. As an example, Khoury [79] notes that INEMA adopts the Q90 as reference flow rate (i.e., 80% of Q90 can be granted for different water uses) based on an outdated time series from 2006. This in itself calls for a territorial strategy to implement a new Corrente River basin plan, which includes not only a hydrological monitoring system [17], but also a review of water grants based on updated hydrological data and periods of lower flow (dry season), as well as assessment of the cumulative systemic effects of irrigation (especially in recharge areas), and a more effective participation process.

*4.3. Hydrological Cycles and Large-Scale Irrigation*

Our results indicate that agricultural change rather than climate change may be the main driver of downward streamflow trends in the western Bahia region. We emphasize the importance of this research finding to understand the water conflicts in the plantations' downstream areas. It should be noted that more than 15 million people inhabit the São Francisco River basin, which encompass the Pratudão River sub-basin. They depend on the river for drinking, producing, etc. [80]. However, the scientific debate over the causes of the water crisis corresponds yet to a local "water controversy", in which soybean producers and smallholders have different perceptions of the water scarcity issue. Since

water pumping takes place in the catchment areas situated in the flat uplands dedicated to soybean cultivation, and water scarcity manifests in distant lowlands located downstream, the link between agribusiness expansion and water depletion is hard to establish in a context of climatic variability. While soybean producers blame rainfall reductions and traditional pastoral management in the valleys for water scarcity, smallholder communities mainly blame large-scale deforestation and irrigation in catchment areas. However, despite very little deforestation in these valleys, there is water scarcity in these areas, even 50 km distant from the plantations.

Indeed, water availability fluctuates over time and space, and streamflow within watershed boundaries have often shown dislocated (downstream) effects in terms of quantities and qualities [81]. Water balance variables have shown a complex behavior based on interactions, retroactions, and recursions in their feedback loops that may affect large extents of land distant from the detected water problem [82]. This can explain certain difference in perception of microclimates and macroclimates on a day-to-day basis as well as on an annual basis across different territorial scales.

Recent hydrological data suggest the need to adopt stricter technical criteria for water grants authorizations [80]. In addition, our findings converge towards the idea that increasing environmental degradation, conflicts, and inequalities in the Cerrado result from political choices that favor agribusiness expansion [21,29,83]. Indeed, in our previous studies, we have stressed how the deregulation of environmental rules in Bahia, based on simplified environmental licensing, has facilitated the indiscriminate issuance of water grants and deforestation licenses [84].

In a context of increasing environmental conflicts in MATOPIBA [21,29], the changes in the hydrological cycle associated with the spatial dynamics of agro-industrial frontiers help explain the problems of water governance at a larger scale.

## 5. Conclusions

In the Brazilian Cerrado, the expansion of soybean monocultures can be considered a global phenomenon of water appropriation. In the Corrente River basin, one of the main regions of expansion of the agricultural frontier in western Bahia state, we observed almost 10-fold increase in agricultural area over the last 30 years. Soybean production is highly dependent on irrigation in this transition area to the semi-arid region.

Our analysis of the streamflow and precipitation rates over the last 40 years in the Pratudão watershed (part of Corrente River basin) based on two hydrological methods indicate a decrease in runoff that might reflect changes in land use, instead of solely climatic factors such as reduced precipitation. In addition, most of the rainfall data indicated too low a number of downward trends series for reduced precipitation to be considered a major contribution to streamflow decrease.

In a context of increasing socio-environmental conflicts in MATOPIBA, more research is need to identify the irrigation water volumes used by companies. We also recommend the inclusion of communities in initiatives aimed to identify negative impacts of the irrigation like systematic mapping of the springs' migration processes, and more effective participation of these communities in water governance of the Corrente River basin (e.g., technical and financial support to the basin committee, creation of technical chambers) along with hydrological and environmental education programs aimed to communities and their representatives.

**Author Contributions:** A.L.d.S.: Conceptualization, methodology and formal analysis, writing—original draft preparation, writing—review and editing; visualization. S.A.d.S.: Conceptualization. methodology and formal analysis, writing—review and editing. O.C.F.: methodology and formal analysis, writing—review and editing. Y.B.S.: methodology and formal analysis, writing—review and editing. L.E.: Conceptualization, writing—review and editing, funding acquisition. C.J.S.P.: writing—review and editing, funding acquisition. All authors have read and agreed to the published version of the manuscript.

**Funding:** This research was supported by Coordination for the Improvement of Higher Education Personnel (CAPES) and Agropolis Foundation, through Project "Sociotechnical and institutional innovations for the conservation and enhancement of the Cerrado biome" (Sociobiocerrado), implemented from 2015 to 2017 (#0340/2015), and CNRS through its "International Emerging Actions Project".

**Acknowledgments:** We thank all our interlocutors who collaborated with this study and their institutional partners. We thank in particular our interlocutors from Brejão, Pratudinho and Praia for their kind hospitality during fieldwork. The authors also gratefully acknowledge the two anonymous reviewers whose comments helped improve this manuscript.

**Conflicts of Interest:** The authors declare no conflict of interest.

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
