# Peer review of "Water Appropriation on the Agricultural Frontier in Western Bahia and Its Contribution to Streamflow Reduction: Revisiting the Debate in the Brazilian Cerrado"

_water, doi:10.3390/w13081054_

Round 1

Reviewer 1 Report

Authors have responded well to reviewers. The revised paper is improved and offers contributions to the field.

One minor point is that authors should resolve the confusion p. 4, lines 135-140, regarding when agricultural settlement began in western Bahia. It was the early 1980s, but the paragraph suggests 1990s and mid 1980s. Satellite imagery from 1986 shows large clearings that were initiated a few years earlier.

Author Response

Response in attachment

Reviewer 2 Report

In their revision to Water appropriation on the agricultural frontier in western Bahia and its contribution to streamflow reduction: revisiting the debate in the Brazilian Cerrado, Leme da Silva et al. have improved the explanations of their quantitative arguments improving the overall quality of their manuscript. I continue to have a few important issues with the paper that I recommend be addressed prior to accepting the paper for publication: 1) I continue to find the lopsided sample-size and inclusion of qualitative data to appear biased and subjective. I appreciate that the authors find the opinions of the large-scale farmers important despite having a sample size of interviews in the single digits. I do not think that the argument offered by the authors in this draft (i.e., that these players are important because they have individually done problematic things in their own water use) to be a convincing argument for inclusion. I suggest this argument be removed as it seems more appropriate for inclusion in journalistic or opinion-based writing than publication in a peer-reviewed scientific journal. I would suggest that these data could remain included but that it be emphasized that the sample size is very small so may not be representative of all large-scale farmers. In addition, both small and large-holder residents have a number of misconceptions and seemingly incorrect assumptions about what’s been happening with water and to the climate in this watershed. The majority of both groups thought precipitation had been decreasing, which according to this analysis, is not the case. This brings into question other observations by both groups – they both have a lot of stake in their side of this argument and both appear to not have all of the information that would be helpful to understanding the ecological implications of large-scale agriculture. I would emphasize this point and emphasize what the interviews tell you – what is the value of their inclusion? And, this is only my opinion, I would suggest that more objectivity in this qualitative data reporting would allow the data in this paper to better be incorporated into policy or acted upon with less clear bias against the large-scale producers and would make the data more appropriate to be reported in a scientific publication. 2) There seem to be a few data gaps in terms of the hydrology. For example, one piece of this puzzle that is now mentioned more in this version but isn’t quantified, is the role of eucalyptus plantations in these watersheds. Eucalyptus trees transpire a lot of water, potentially at rates well beyond background ET rates in the Cerrado. This may itself also contribute to streamflow decreases over this time period. This may be unlikely to contribute as much as water drawdown from irrigation, but a look at ET rates would further bolster the case for a reduction in streamflow from land use and agriculture rather than a change in precipitation. 3) There also continues to be a portion of the conclusion focused on the “fact” that this region has gotten drier. To me this implies a decrease in ppt, which the authors argue is not happening. This deserves further explanation. Do the authors mean ppt or something else?

Minor comments:

The paper uses variable significant figures in the reported numbers. I suggest consistency.

Line 31: What percentage of farmers? 100%? Please report if you are quantifying the smallholder responses here.

Line 114: Is the São Francisco basin really the most important of Brazil’s river systems? What about the Amazon?

Lines 296-301: See comments above. I would remove this discussion of the individual bad actors you are describing. Is there a reason to include this small sample size that is not so overtly political and demonizing of one group?

Line 321: Minus 10%? Could this just be written as -10%?

Figure 2: I continue to find your phrasing here confusing. When you say “no significant downward trend” you’re actually saying: there was a downward trend but it wasn’t significant, it seems. The data in d and e are interesting and suggest something might be at play here in terms of ppt: part of the watershed is getting significantly wetter while part of it is getting significantly drier – while this isn’t as pronounced as streamflow are there other possibilities for what might be happening?

Tables 3 and 4: Please write out in full what the various acronyms and abbreviations mean.

I don’t understand the points in lines 486-495 and lines 486-506 – what do you mean short and long term persistence? Short- and long-term persistence of what? The presence of what feature increases the type I error? Do you mean the chance of type I error? Please clarify your points in these paragraphs to distinguish between your study and the study described here.

Line 514: What is FDC? I don’t see this acronym defined previously

Line 519: An important piece of additional evidence relates to the eucalyptus planted in the headwaters – this would greatly increase evapotranspiration in this part of the region potentially leading to reductions in streamflow as well. How extensive are these plantations?

Author Response

Please find response in attachment

Round 2

Reviewer 2 Report

Thank you for responding to my concerns. I have now recommended that your paper be accepted for publication.

Author Response

Dear Reviewer 2, 

I am grateful for your valuable suggestions provided in the review process. Please find minor revisions highlighted in the attached manuscript.

Best regards,
Andrea Leme Silva

This manuscript is a resubmission of an earlier submission. The following is a list of the peer review reports and author responses from that submission.

Round 1

Reviewer 1 Report

The paper focuses on an important problem in the Cerrado region of Brazil, a globally significant agricultural producing region, where for many years the potential impact of increasing irrigation demands on surface and groundwater has been poorly understood. The November 2017 conflict shows the high relevance of this problem and urgent need for hydrological and social understandings of this problem. This paper makes a significant contribution to this issue.

Lines 80-2: could the authors offer more precise and clear corrections or rebuttal to the previous work cited in regard to western Bahia (Gaspar and Campos; Pousa et al)? The critique offered line 457 is unclear and could be clarified and detailed. (The subtitle of the paper suggest that a debate will be revisited, so readers should have greater clarity on this debate and what the present paper adds to it.)

Lines 126-7: here the authors blend land covers (veredas) with land uses and tenure regimes (fecho de pasto, quilombolas, geraizeros). More detailed explanation is important here.

Lines 235-: The November 2017 conflict likely has influenced the data collection in communities. More reflection on this is required.

Lines 252-: could the authors offer stronger justification for selection of the Pratudão watershed, and additional context for the claim for representativeness of western Bahia? Other areas with irrigation in western Bahia produce coffee and fruit; and in other areas in western Bahia, farmers extract water from the aquifer. Is irrigation here based on river pumps? (Lines 347-50 offer details that could have been made clearer earlier in the paper.) Are water demands for seed production greater or less than water demands for coffee and fruit?

Section 3.1: did the analysis reveal increased stream “flashiness,” that is, was higher peak streamflow observed even as discharge decreased over time?

Line 326: “correlating” seems to be used inappropriately here to describe how the authors dealt with the land cover datasets.

Lines 363-384: could the authors provide more information on the ecological indicators of drying? And (line 375) on the spring migration phenomenon? I think the Conclusion of the paper should suggest to readers how future research might investigate or validate these processes.

Lines 395-408: the views of soy producers documented in this paper seem to reproduce views of “agri-environmentalists” who viewed the natural vocation of the Cerrado in similar ways as documented in earlier work, https://doi.org/10.1080/00330124.2011.585081.

Lines 487-90: could the authors offer more reflection on how their findings compare with other findings in the Cerrado? Are similar methods and data types used? The paragraph 509-512 seems to be best suited as the start of a longer and more detailed paragraph on how the present studies compares or adds to previous studies of stream runoff declines in the Cerrado.

Lines 551-52: could the authors offer more direct engagement with Favareto 2019 and Porto-Gonçalves 2019? Should we reconsider environmental conflicts in this region based on the present paper’s findings? Specific engagement with the activist literature is important here.

Lines 580-2: what recommendations or suggestions are there for communities? Are there citizen science initiatives that might help document the ecological and hydrological changes? Other areas of conflict, such as highland Andes downstream of mining sites, well described by Bebbington in https://doi.org/10.1073/pnas.0906057106, offer recommendations for the inclusion of communities in initiatives that create stronger data basis for discussion of negative impacts. It also seems as though technical criteria for water grants are not enough; the water bureaucracy seems to entirely lack monitoring of water actually taken from streams and applied on crops. It is also possible that they lack understanding of the return of applied irrigation water to surface flow.

Reviewer 2 Report

In their paper, Water appropriation on the agricultural frontier in western Bahia and its contribution to streamflow reduction: revisiting the debate in the Brazilian Cerrado, Leme da Silva et al. use publicly available datasets and a series of interviews with small and large-scale farmer to assess trends in hydrology (streamflow and precipitation) in watersheds in the Cerrado biome of Brazil. The Cerrado is a biodiversity hotspot and a sensitive ecosystem that has seen rapid and extensive conversion to agriculture over the past several decades. The paper’s main position is that it is agricultural drawdown of groundwater resources for irrigation and not climate change that is responsible for the observed reduction in streamflow over time in the Corrente River basin. The interplay between decreased evapotranspiration following the clearing of natural vegetation, loss of groundwater from irrigation, and climate change is such an important question for an unquestionably important ecosystem. However, the authors study is very similar to a recent study (Pousa et al. 2019), also published in Water and looking at the same ecosystem, but with shorter time series and less well-analyzed datasets. The authors results contrast in part with this other study regarding trends in precipitation over time and this seems to be what the paper is actually wanting to communicate. However, they never made this explicit nor was it clear to me that this study’s results were more robust than the other. The other novel information presented here is the results of interviews with local farmers. There was no quantification or objective reporting of the interview results, making it difficult to know whether these results were reported without bias, particularly as the bias expressed in the reported results support the author’s major claims. I don’t doubt that drawdown of groundwater resources is having a major impact on streamflow and water conflicts in this region but the authors have either reported data that have been reported elsewhere or data that do not fully support their claims.

Major Comments:

Line 145: What do you mean data from nearby sites? Do you mean that you were only looking at the Pratudão watershed but analyzed all sites from the Corrente watershed? Why not just say you looked at the Corrente watershed then? It looks like it is because there are no flow stations in your chosen watershed. Particularly because you are considering headwater watersheds, you should definitely include flow data for the watershed in question.

Lines 156-161: I’ve never seen “advanced visual examination of the data” used to describe quantitative data analysis. I think you can just report what you did: you calculated the mean, maximum, and minimums ppt and streamflow, presumably over time, etc. How did you account for the discovered data problems? What did you do to fill gaps, for example?

Lines 162 – 171: Could you provide more information here? What do you mean “drainage area ratio transfer”? You also say that you used the series of streamflow in the Prutudão watershed, but you don’t have any data from there. Do you mean the station 45840000 data?

Line 210-227: The TFPW can obscure some trends that actually are significant. There are some suggested modifications in the literature. Perhaps consider one of these to ensure you aren’t missing trends that actually are present, particularly as your data contrast with multiple other studies looking at the same types of trends.

Line 233: Interviews: why were there more than twice as many interviews with small vs. large holding farmers?

Figure 2: It remains very unclear what data you are comparing here: there are no fluviometric stations in the Pratudão River Basin – is this looking just at the one closest sampling station or are you looking across all stations in the Corrente watershed? If it’s all, it should definitely be labeled as such. Maximum and Minimum flow rates aren’t particularly relevant data points here – It would be more interested in total export from the watershed, for example, or total baseflow each year, or if you do want to report max and min, consider the max and min mean daily flow, as this is more representative of a time period rather than just one moment of flow.

Figure 3: What are the units for streamflow and rainfall? Is is mm/day for both? You haven’t previously reported flow corrected by watershed area, but that’s what I would expect for runoff coefficient.

Lines 366 and later in lines 489-493: The effect of deforestation on ET is, in most cases, including Spera et al. 2016 which you use as evidence of the opposite, is that ET decreases leading to increased streamflow. This is the pattern of regional drying and climate change that we expect: clearing the forest leads to larger exports of local water such that less is recycled and over time the local climate dries out leading to an ecosystem that can no longer maintain itself after some level of clearing. This is really important and even if this isn’t what is happening in this watershed at this time it certainly is happening in the Amazon Basin and is a very dangerous trend and should be described clearly.

Lines 357 – 408: There is some quantification of responses from small-holder farmers, but no objective analysis of the soybean producer interviews. How many people mentioned the phrases you mention? Did any have similar answers across groups? As reported it isn’t clear that you have been objective in your analysis of these interviews.

Lines 428 – 438: The fact that your results contradict those of eight other studies directly in terms of precipitation trends deserves much more attention. What makes your analysis of available rainfall data any better than theirs? What novel way have you approached these data to come up with a different result? I don’t doubt that groundwater drawdown might be currently playing a larger role than regional climatic change in streamflow reductions, but I do not think that you have shown that here with your analyses. In Lines 453-455 you say “It is a fact that the last two decades have been drier in Corrente basin than earlier” but you, in fact, are arguing the exact opposite throughout the paper!

Minor Comments:

Line 51-52: It may be true in the Cerrado that soybean expansion has been facilitated through irrigation, but in the Amazon forest ecosystem soybeans have also rapidly and extensively expanded through the development of varieties that can tolerate humid tropical climates and extensive soil modifications to facilitate growth on latosols.

Line 98: I think you could explain what you mean by “mixed quali/quanti database” here. That language isn’t common parlance and I did not understand what you meant. Perhaps you could take a moment to describe the data in more detail.

Line 149-150: You don’t need to describe the details of the figure legend in the text.

Figure 1: What do you mean “with 45840000 flow series”?

Figure 4: It’s confusing to read “no trend (downward)”, etc. Perhaps find a different way to say that no significant trend was detected. You also don’t report p-values for the trends that aren’t significant – how close are they to 0.05?